# *Limosilactobacillus reuteri* Strains as Adjuvants in the Management of *Helicobacter pylori* Infection

**DOI:** 10.3390/medicina57070733

**Published:** 2021-07-20

**Authors:** Costantino Dargenio, Vanessa Nadia Dargenio, Francesca Bizzoco, Flavia Indrio, Ruggiero Francavilla, Fernanda Cristofori

**Affiliations:** 1Interdisciplinary Department of Medicine, Pediatric Section, University of Bari, Children’s Hospital “Giovanni XXIII”, 70126 Bari, Italy; costy.dargenio@gmail.com (C.D.); vanessa.dargenio@unifg.it (V.N.D.); francescabizzoco.med@gmail.com (F.B.); fernandacristofori@gmail.com (F.C.); 2Department of Medical and Surgical Sciences, Pediatric Section, University of Foggia, 71122 Foggia, Italy; flaviaindrio1@gmail.com

**Keywords:** *Helicobacter pylori*, probiotics, *Lactobacillus reuteri*, therapy, resistance

## Abstract

*Helicobacter pylori* (HP) is a Gram-negative bacterium which finds its suitable habitat in the stomach. The infection affects about half of the global population with high variability in prevalence among regions and for age. HP is the main causative agent of chronic active gastritis, peptic and duodenal ulcers, and may be the primary cause of gastric cancer or MALT lymphoma. Due to the high rate of failure of eradication therapy in various countries and the increase in antibiotic resistance reported in the literature, there is an ever wider need to seek alternative therapeutic treatments. Probiotics seem to be a promising solution. In particular, the *Limosilactobacillus reuteri* (*L. reuteri*) species is a Gram-positive bacterium and is commonly found in the microbiota of mammals. *L. reuteri* is able to survive the gastric acid environment and bile and to colonize the gastric mucosa. This species is able to inhibit the growth of several pathogenic bacteria through different mechanisms, keeping the homeostasis of the microbiota. In particular, it is able to secrete reuterin and reutericycline, substances that exhibit antimicrobial properties, among other molecules. Through the secretion of these and the formation of the biofilm, it has been found to strongly inhibit the growth of HP and, at higher concentrations, to kill it. Moreover, it reduces the expression of HP virulence factors. In clinical trials, *L. reuteri* has been shown to decrease HP load when used as a single treatment, but has not achieved statistical significance in curing infected patients. As an adjuvant of standard regimens with antibiotics and pump inhibitors, *L. reuteri* can be used not only to improve cure rates, but especially to decrease gastrointestinal symptoms, which are a common cause of lack of compliance and interruption of therapy, leading to new antibiotic resistance.

## 1. Introduction

*Helicobacter pylori* (HP) is a Gram-negative flagellated spiraliform bacterium about 3 micrometers in size which finds its suitable habitat in an acidic environment. It survives in a poorly oxygenated atmosphere, and the ideal environment is represented by the gastric mucus present in the stomach [1]. HP affects more than half of the world population, with a wide variability among different regions, going from 79.1% in Africa to 50% in Southern Europe in 2015, despite the fact that the prevalence of *H. pylori* has been rapidly declining for decades in some parts of the world [2]. There is also a strong variation in prevalence between individuals over the age of 60 (50%) and those under the age of 40 (20%) worldwide, with the latter being at lower risk of colonization [3].

HP is the main causative agent of chronic active gastritis and plays a leading role in peptic and duodenal ulcers, although patients may often be asymptomatic. The infection may be the primary cause of gastric cancer or MALT lymphoma; the risk of cancer is not related to the severity of infection, but to the extent of atrophy with the development of atrophic gastritis involved in the gastric body [4,5,6]. Treatment of HP infection is based on pump inhibitors (PPIs), which neutralize the acidic environment, and two or more antibiotics and eradication must be confirmed two months after treatment. Bismuth quadruple therapy is the best empiric regimen in areas with high rates of antibiotic resistance, in patients who have previously used macrolides or quinolones, or after failure of clarithromycin or levofloxacin regimens. Triple therapy with clarithromycin is still recommended in geographic areas with a lower likelihood of a clarithromycin-resistant strain. However, due to the high rate of failure of eradication therapy reported in the literature in various countries [7] and the increase in antibiotic resistance [8], there is an ever wider need to seek alternative therapeutic treatments. According to the Maastricht V Consensus Report, an adjuvant therapy could be represented by specific probiotic species, including *Lactobacillus reuteri species*, recently renamed *Limosilactobacillus* (*L.*) *reuteri*, and a challenge has arisen in identifying which probiotic may be the most appropriate [9]. Probiotics are live microorganisms capable of providing health benefits when consumed, generally by improving or restoring the gut microbiota [10], and are generally considered safe to consume. Only in rare cases can they cause bacterial–host interactions and undesirable side effects [11].

*L. reuteri* species is a part of the Gram-positive non-sporulating bacteria that include several strains and are commonly found in the microbiota of mammals and in the fermentation of several foods [12]. As a probiotic, it is a very resilient species; it is able to survive the gastric acid environment and bile and to colonize the gastric mucosa [13]. The *L. reuteri* species is able to inhibit the growth of several pathogenic bacteria through different mechanisms. In particular, this species is able to secrete 3-hydroxy propionaldehyde, also called reuterin [14,15], a substance that exhibits broad-spectrum antimicrobial properties, and reutericycline, a defense molecule especially against Gram-positive bacteria. Reuterin has been found to strongly inhibit the growth of HP and, at higher concentrations, to kill it. Moreover, it reduces the expression of HP virulence factors such as the products of VacA and flaA genes [16]. Finally, *L. reuteri* DSM 17938 can secrete microvesicles and anti-agent adhesion complex molecules such as exopolysaccharides that can inhibit the adhesion of different bacteria (i.e., *Escherichia coli*) [17]. Although the mechanism of action of microvesicles is still under study, it is known to serve for the interaction with gut cells and other bacteria [18,19] and to produce a biofilm which protects its environment and, at the same time, mucosal cells, allowing their survival [20,21].

In one of the first studies assessing the role of probiotics in HP infection, Midolo et al. showed that lactic acid produced by Lactobacilli could inhibit the growth of HP at concentrations of 50–156 mmol/L [22]. Mukai et al. identified some receptors which are contended between HP and some *L. reuteri* strains for the adhesion to the gastric mucosa. In particular, some of these strains could bind to the glycolipids asialo-GM1 and/or sulfatide in vitro, possibly hindering the adhesion of HP [23]. In an in-animal study, Zaman et al. discovered that three lactobacillus species (*L. reuteri*, *L. johnsonii* and *L. murinus*) residing in the gastric mucosa of some Mongolian gerbils exhibited an inhibitory effect on the in vitro growth of HP. Moreover, the colonization rate of HP in gerbils hosting these three strains was lower than the rate in gerbils without HP (67% vs. 40%) [24].

Given these very specific features, *L. reuteri* has been tested as a possible adjuvant in therapy for the eradication of HP. In particular, the activity of *L. reuteri* was tested alone or with antibiotics.

## 2. Use of *L. reuteri* 17938 (Or Mother Strain ATCC 55730) Against HP Infection without Antibiotics

Saggioro et al. were among the first to use *L. reuteri* ATCC 55730 (the mother strain of DSM 17938) at the dose of 10^8^ CFU, twice daily in a randomized trial. Thirty patients were randomized to receive omeprazole plus *L. reuteri* ATCC 55730 (10^8^ CFU, twice daily) or plus placebo, for 30 days. The eradication rate was 60% in patients supplemented with *L. reuteri*, as compared to none in the group that received omeprazole plus placebo (*p* < 0.0001) [25].

Imase et al. showed that *L. reuteri* ATCC 55730 at the daily dose of 10^8^ CFUs could significantly decrease the 13-C-urea breath test (13-C-UBT) delta-value by 70% as compared to placebo (*p* < 0.05), showing that *L. reuteri* is able to decrease HP bacterial load at the gastric level [26].

Francavilla et al., in a double-blind placebo-controlled study, randomized 40 HP-positive subjects to receive daily *L. reuteri* ATCC 55730 or placebo for 4 weeks. All underwent 13-C-UBT, and HP stool antigen determination at entry and at the end of treatment. The authors were able to show that *L. reuteri* reduces HP load as semi-quantitatively assessed by both the 13-C-UBT delta value and HP stool antigen quantification after 4 weeks of treatment (*p* < *0*.05). Moreover, *L. reuteri* but not the placebo was followed by a significant decrease in the Gastrointestinal Symptom Rating Scale (GSRS) as compared to pre-treatment values (*p* < *0*.05) [27].

In an open-label pilot study, Dore et al. administered *L. reuteri* DSM 17938 10^8^ CFU per dose plus pantoprazole 20 mg twice a day for 60 days in 22 adults. This combination treatment achieved HP eradication in 13.6% (3/22; 95% CI 2.9–34.9%) for intention-to-treat analysis and in 14.2% (3/21; 95% CI 3.0–36%) for per-protocol analysis; moreover, the authors reported a decrease in urease activity of 76%. The main limitation of this study is the absence of a control group, and it cannot be excluded that these results might be secondary to the 60-day therapy with PPI [28].

In a two-site randomized study by Dore et al., 53 HP-infected subjects were administered pantoprazole and a mixture of two *L. reuteri* strains (Gastrus^®^: 2 × 10^8^ CFU *L. reuteri* DSM 17938 plus 2 × 10^8^ CFU *L. reuteri* ATCC PTA 6475) or placebo seven times daily for four weeks. The difference in cure rate was 12.5% in the active group compared to 4.1% in the placebo group; the results, although promising, were non-significant [29].

## 3. Use of *L. Reuteri* 17938 (Or Mother Strain ATCC 55730) Against HP Infection without Antibiotics

Scaccianoce et al. enrolled 65 consecutive dyspeptic patients with HP infection to receive the standard 7-day triple plus *L. reuteri* (n: 16) or a probiotic mixture (n: 17) or placebo (n: 15). Eradication rate was not different in the three groups; however, the lowest incidence of antibiotic-associated side effects was observed following *L. reuteri* administration [30].

Lionetti et al. randomized 40 HP-positive children to receive sequential therapy plus *L. reuteri* ATCC 55730 (10^8^ CFU) or placebo. Overall, all children supplemented with the probiotic as compared to those receiving placebo reported a significant reduction in GSRS score during eradication therapy and at the end of follow-up. No difference was found in the eradication rate [31].

*L. reuteri* was administered as an adjuvant of a second-line triple therapy with levofloxacine in a randomized trial by Ojetti et al., resulting in a statistically significant increase in eradication rate (80% in the probiotic group vs. 62% in the control one; *p* < 0.05) and a significant decrease in nausea and diarrheal episodes, even of moderate to severe intensity (*p* < 0.05) [32].

Efrati et al. conducted a trial to compare standard triple therapy versus sequential regimen for HP eradication. All 83 patients irrespective of the treatment regimen were given *L. reuteri* either during or after antibiotics. The sequential treatment showed a significantly higher eradication rate of HP compared with the standard regimen and the authors demonstrated an extraordinarily low incidence of antibiotic-associated side-effects and a very good compliance, probably related to the use of *L. reuteri* supplementation [33]. 

Our group performed a double-blind, randomized, placebo-controlled trial to evaluate the efficacy of two *L. reuteri* strains (Gastrus^®^). Probiotics were given to 100 HP-positive naive patients during pre-eradication (28 days before antibiotics), eradication (standard triple therapy), and follow-up (8 weeks). We showed that this specific combination was able to significantly decrease HP bacterial load (pre-treatment), significantly reduce antibiotic-related side effects (41% vs. 63%; *p* < 0.04), reduce serum gastrin-17 (28% vs. 12%; *p* < 0.02) (treatment) and increase the eradication rate by 9% (follow-up) compared to placebo [34].

In a similar randomized trial by Emara et al., the patients were randomized to standard triple therapy for two weeks, plus Gastrus^®^ for 4 weeks. The authors found a 9% higher eradication rate and a 30% decrease in antibiotic-related side effects (diarrhea, abdominal distension and taste changes) in patients receiving Gastrus^®^ as compared to placebo. Interestingly, the authors were able to repeat endoscopy after treatment and demonstrate that the gastric inflammatory cell score and the activity score were significantly reduced in the probiotic group as compared to the control [35].

In a case series, Dore et al. studied a possible replacement of bismuth with the probiotic *L. reuteri* (10*8 cfu once a day for 20 days) in the quadruple therapy with pantoprazole, tetracycline and metronidazole. This combination was very successful, with an eradication rate of 93% and few side effects reported with optimal compliance [36]. The same group completed a randomized study with standard quadruple therapy with either bismuth or Gastrus^®^ once daily. The eradication rate was 100% in the bismuth-containing regimen against 90% in the probiotic group. The authors concluded that Gastrus^®^ might be considered when using bismuth is contraindicated or unavailable [37].

Poonyam et al. used Gastrus^®^ or placebo in combination with a quadruple therapy with bismuth in a randomized trial. At seven days, no difference in the percentage of cure rate can be found (68% probiotic vs. 72% placebo), but at 14 days, a trend can be seen even if still non-significant in favor of the probiotic regimen (96% probiotic vs. 88% placebo), independently from antibiotic resistance or patient metabolism of these drugs. Finally, common side effects such as nausea, vomiting, abdominal discomfort, and bitter taste were significantly reduced in the probiotic regimen [38].

In a recent randomized trial on quadruple therapy with bismuth plus Gastrus^®^ or placebo, Moreno Marquez et al. did not find any difference in the eradication rate between the two regimes in the 80 patients enrolled. However, patients receiving Gastrus^®^ reported significantly less abdominal pain and abdominal distension compared to the control group [39].

## 4. Use of Different Strains Rather Than *L. reuteri* 17938 Against HP Infection without Antibiotics

Aside from live bacteria, Mehling et al. administered Pylopass, a compound of spray-dried *L. reuteri* DSMZ 17648 cultures, to HP-positive patients, resulting in a decrease in 13-C-UBT delta values, which was maintained even after 24 weeks. This confirms the presence of molecules secreted by *L. reuteri* or its components, which act on the infective agent independently of its probiotic activity [40].

In another study, *L. reuteri* DSM 17648 was found to be able to co-aggregate with HP in vitro and in vivo. This specific co-aggregation (with pathogen blocking abilities) occurs between *L. reuteri* DSM17648 and different HP strains such as *Helicobacter heilmannii*. This *L. reuteri* strain was shown to significantly decrease the HP bacterial load in infected adults in two weeks, suggesting that it might be used as an adhesion blocker in anti-HP therapies [41]. In a randomized trial in 46 patients, Muresan et al. showed that triple therapy with Pantoprazole for 30 days associated with Amoxicillin, and Clarithromycin for 14 days, was comparable with a regimen of Pantoprazole plus *L. reuteri* DSMZ 17648 twice a day for 8 weeks with a better trend in favor of the antibiotic group as compared to the probiotic group (74% vs. 65%, respectively) [42]. Buckley et al. concluded that *L. reuteri* DSM 17648 reduces *H. pylori* load, improving gastrointestinal symptoms in subjects positive to this infection, and improving antibiotic–gastrointestinal adverse effects that are common after eradication treatment with antibiotics [43].

In a meta-analysis conducted by Shi et al., including 40 studies with 8924 patients, the efficacy of different probiotics in facilitating the eradication of HP and reducing side effects during the eradication therapy was evaluated. Supplementation with probiotics before and throughout the antibiotic treatment, and also, the use of probiotics for more than 2 weeks, exerted better eradication effects. *Lactobacillus* and multiple strains were the best choices for probiotic strains, although species were not specified. The eradication effects of probiotics with the quadruple bismuth regimen were the best combination [44].

Instead, Yu et al. carried out a meta-analysis of randomized controlled trials of *Lactobacillus* supplementation on HP eradication rates and antibiotic-associated side effects. Eleven randomized controlled trials involving a total of 724 patients were included. The authors found that the HP eradication rate in the *Lactobacillus* supplementation group was significantly higher than in the control group (RR 1.16; *p* < 0.0001), regardless of age (adults or children) and geographic location of the study (Asia or Europe). Interestingly, in the subgroup analysis based on *Lactobacillus* strains, eradication rates were significantly increased in the *L. reuteri* and *L. casei* groups, whereas it seems that *Lactobacillus* GG could not improve eradication rates. In addition, *Lactobacillus* supplementation significantly reduced antibiotic-associated side effects (RR = 0.36, 95% CI 0.17–0.74, *p* = 0.005) [45].

None of the reviewed studies reported side effects from the probiotic.

## 5. Conclusions

In conclusion, data available for the analysis give us a perspective of the possible use of the probiotic. *L. reuteri* as an adjuvant treatment in the eradication therapy of HP. *L. reuteri* meets all the criteria necessary for a microbial strain to be considered safe and intended for human use such as: (a) human origin, (b) being isolated from the gastrointestinal tract of healthy subjects as a non-pathogenic strain, (c) ability to survive gastric acidity and bile, (d) colonization of the human intestine, (e) not carrying transmissible genes for antibiotic resistance, and (f) scientifically proven clinical efficacy [46]. Many clinical trials have demonstrated the safety, efficacy, and tolerance of *L. reuteri* in the treatment of *H. pylori* infection; however, several limitations are present. First, in the different trials, there was diversity in the antibiotics used in the different therapies, the timing for confirmation of *H. pylori* eradication, and the timing of probiotic administration, which could vary the effect size of probiotics between studies. In addition, most of the trial designs were different among the studies performed [47]. Overall, the concomitant use of *L. reuteri* with eradication treatment (independent of the regimen used) can increase the eradication rate by about 10% [48]. Moreover, probiotics are able to decrease antibiotic-associated side effects by about 20–30% [44]. In general, all probiotics, including *L. reuteri,* cannot substitute the antibiotic treatment that remains the gold standard treatment for this infection although, some strains that have been proven efficacious in RCTs (L.R 17938 or PTA6475) can decrease HP bacterial load. 

In conclusion, some strain of *L. reuteri* can be used in combination to antibiotics not only to improve cure rates, but especially to decrease gastrointestinal symptoms which are a common cause of lack of compliance and interruption of the therapy which lead to new antibiotic resistance.

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
