# Peer review of "Limosilactobacillus reuteri Strains as Adjuvants in the Management of Helicobacter pylori Infection"

_medicina, 2021, doi:10.3390/medicina57070733_

Round 1

Reviewer 1 Report

Dargenio et al. comprehensively summarized known evidence about the use of Limosilactobacillus reuteri strains as adjuvants in the management of Helicobacter pylori (HP-)-infection. 

Overall, this review is clearly structured and very well written. The reader is nicely guided through the existing literature and results together giving a clear and informative overview.

Albeit, for achieving standards of a proper review of the literature, in its current version the manuscript is too much of enumerative nature, meaning that there is missing more background information and integration of data within the broader context.

Major points:

  • the authors should add more background information: 
    • definition and context information on probiotics in general
    • more precise information regarding current guideline suggestions for the treatment of HP => need of new therapeutic options regarding increasing antibiotic resistance rates, etc.
    • rationale and general considerations / limitations in the use of probiotics (=> reference to limited but existing data in the cochrane database; relevant context of ameliorating antimicrobial resistance; etc.)
  • the authors should add integration of the shown evidence in the context HP-eradication
  • if existing; the authors should give an overview (table) of all/most important commercially available Limosilactobacillus reuteri formulations (concentration, etc.)
  • the authors should point out differences / commonalities of Lactobacillus reuteri and Limosilactobacillus reuteri strains and respective interactions, respectively

Minor points:

  • line 191: please add reference for this statement => is really “postbiotics” meant? If so, this term should be specified
  • line 202 - 205: “Overall, the concomitant use of L. reuteri with eradication treatment (independently of the regimen used) can increase the eradication rate of about 10%. Moreover, probiotics are able to decrease antibiotic associated side effects of about 20-30%.”: please add respective references

Author Response

Major points:

the authors should add more background information:

1.definition and context information on probiotics in general

Answer

Line 54. The general introduction has been modified as suggested.

2.more precise information regarding current guideline suggestions for the treatment of HP => need of new therapeutic options regarding increasing antibiotic resistance rates, etc.

Answer

Line 49-53 the text has been expanded as indicated by the reviewer's comments.

3.rationale and general considerations / limitations in the use of probiotics (=> reference to limited but existing data in the cochrane database; relevant context of ameliorating antimicrobial resistance; etc.)

Answer

Line 233-245. In the final part of the elaboration the considerations suggested have been inserted.

4.the authors should add integration of the shown evidence in the context HP-eradication

Answer

These changes have been incorporated in the text as requested by the reviewer.

5.if existing; the authors should give an overview (table) of all/most important commercially

available Limosilactobacillus reuteri formulations (concentration, etc.)

Answer

In the studies cited usually the formulations were provided by the manufacturer with already precise CFUs. This request is unfortunately not exhaustible due to differences in commercial legislation from country to country.

6.the authors should point out differences / commonalities of Lactobacillus reuteri and

Limosilactobacillus reuteri strains and respective interactions, respectively

Answer

At the line 59 is the new taxonomy and from line 59 to 90 the functions of L. reuteri species are listed and described.

Minor points:

7.line 191: please add reference for this statement => is really “postbiotics” meant? If so, this term should be specified

Answer

The term postbiotic was deleted where it was named in the text.

8.line 202 - 205: “Overall, the concomitant use of L. reuteri with eradication treatment (independently of the regimen used) can increase the eradication rate of about 10%. Moreover, probiotics are able to decrease antibiotic associated side effects of about 20-30%.”: please add respective references

Answer

Thank for this comment, the references have been added.

Reviewer 2 Report

This is an interesting review written by researches with competence within the field and who have contributed with original research. I have the following remarks and suggestions to improve the manuscript:

Line 39-41: Prevalence of H pylori. Infection normally occurs in childhood and the prevalence of H pylori is low in children in Western populations and figures lower than 32.6% should be quoted to reflect the range. As the prevalence of H pylori has been rapidly declining for decades, this is relevant.

Line 43-44: Please rephrase “in severe cases” as the risk of cancer is not related to severity of infection per se, but to extent of atrophy with development of atrophic gastritis involving the gastric body.

Line 49: Please refer to the more recent Maastricht V consensus from 2016 on the role of probiotics.

Line 192: Conclusions. The results of the meta-analysis (Yu et al) should not be introduced in the Conclusions for the first time, please change this. Furthermore, the majority of the studies included in the meta-analysis (Yu et al) did not involve L reuteri and in the sub-analysis including studies of Lactobacillus GG, did not find increased H pylori eradication rates. However, the studies L ruteri did and this may be of importance. Please delete the statement “irrespective of Lactobacillus genera”.

Author Response

1.Line 39-41: Prevalence of H pylori. Infection normally occurs in childhood and the prevalence of H pylori is low in children in Western populations and figures lower than 32.6% should be quoted to reflect the range. As the prevalence of H pylori has been rapidly declining for decades, this is relevant.

Answer

Line 39-42. The sentence has been modified as suggested by the reviewer.

2.Line 43-44: Please rephrase “in severe cases” as the risk of cancer is not related to severity of infection per se, but to extent of atrophy with development of atrophic gastritis involving the gastric body.

Answer

Line 46-48. Thank you for your suggestion. the sentence in the text has been modified as is needed.

3.Line 49: Please refer to the more recent Maastricht V consensus from 2016 on the role of probiotics.

Answer

Thank you. both the text and the reference have been changed

4.Line 192: Conclusions. The results of the meta-analysis (Yu et al) should not be introduced in the Conclusions for the first time, please change this. Furthermore, the majority of the studies included in the meta-analysis (Yu et al) did not involve L reuteri and in the sub-analysis including studies of Lactobacillus GG, did not find increased H pylori eradication rates. However, the studies L ruteri did and this may be of importance. Please delete the statement “irrespective of Lactobacillus genera”.

Answer

Thanks to the reviewer for the comment. the text has been edited as suggested. Line 212-221.

“Instead, Yu et al carried out a meta-analysis of randomized controlled trials of Lactobacillus supplementation on HP eradication rates and antibiotic-associated side effects. Eleven randomized controlled trials involving a total of 724 patients were included. The authors found that the HP eradication rate in the Lactobacillus supplementation group was significantly higher than in the control group (RR 1.16; p<0.0001) regardless of age (adults or children), geographic location of the study (Asia or Europe). Interestingly, in the subgroup analysis based on Lactobacillus strains, eradication rates were significantly increased in the L. reuteri and L. casei groups, whereas it seems that Lactobacillus GG could not improve eradication rates. In addition, Lactobacillus supplementation significantly reduced antibiotic-associated side effects (RR = 0.36, 95% CI 0.17-0.74, P = 0.005)”